# Optimal Design of a Canopy Using Parametric Structural Design and a Genetic Algorithm

Saaranya Kumar Dasari [1], Nicholas Fantuzzi [2,*], Patrizia Trovalusci [1], Roberto Panei [3] and Marco Pingaro [1]

1    DISG Department, Sapienza University of Rome, 00184 Rome, Italy
2    DICAM Department, University of Bologna, 40126 Bologna, Italy
3    Municipal Urban Hygiene Company (AMA), 00142 Rome, Italy
*    Correspondence: nicholas.fantuzzi@unibo.it

**Abstract:** The structural performance of any building design is often dependent on the geometrical shape, which affects its behavior and stability. Structural consideration and optimization in the conceptual stage of the design process can lead to better solutions and design exploration. In this paper, a design approach for generating and structurally optimizing the geometrical form in the conceptual design phase is presented. The method is applied to a canopy of an ecological island (waste collection center in Rome, Italy). We demonstrate how parametric structural design can facilitate the decision-maker to generate and analyze the optimal design solutions rapidly in the conceptual stage of the design process. Fully parametric models are created in a Rhinoceros3D® environment and interfaced with in-house built algorithms, and Finite Element simulations are performed in DubalRFEM. An ecological island's canopy has been completely redesigned with a Genetic Algorithm and a Dynamic Relaxation Algorithm, resulting in a free-form shape-resistant structure. Finally, the shape-optimized canopy meets various requirements (structural, functional, formal) that improve structural efficiency and design collaboration, such as in the role of the architect and engineer in the design process and in the relationship between the designer and design tools.

**Keywords:** structural optimization; conceptual structural design; genetic algorithms; dynamic relaxation; finite element analysis

## 1. Introduction

The role of optimization has been increasing and is considered an effective approach in every aspect of the engineering field. Everyone is trying to improve the design and engineering process in diverse ways. This fact leads to an increase in demand for low-cost, lightweight constructions. As a result, design optimization is becoming more popular [1]. In the past, architects and engineers have consistently used form-finding and form-improving methods that accurately identify the optimal shapes by using experimental tools and physical models to simulate a particular (expected) mechanical behavior using the reverse hanging, minimal surfaces, and geometric forms, where they have strategies and particular methods for creating these forms [2,3]. Computational techniques have found success over the past few decades in a variety of engineering domains, and sophisticated software now offers a wide range of potential applications [4,5]. Due to this, Finite Element (FE) methods are now necessary tools for all engineering design projects. Numerous optimization algorithms are available in the literature, depending on the optimization problem, that can provide a general approach to automate the structural design process [6–10]. Many of them fall into one of two categories: gradient and heuristic methods. Gradient methods search for an optimum in the design space on the computation of gradient vectors using the derivatives of the objective function [11]. Heuristic methods are established by adopting principles of natural selection. These methods can be established by a set of random variables with probability distributions, which are also known as stochastic optimization

algorithms [12,13]. Researchers are becoming increasingly interested in heuristic search algorithms. An evolutionary algorithm, also known as a genetic algorithm, is a common example of a heuristic method that employs random selection as a tool to search for the most effective solution in the design space. Although, many other methods are available in the literature, such as Simulated Annealing and Particle Swarm Optimization. However, most of these methods were not developed with structural design problems in mind, at least within the construction industry.

The historical developments of shape-resistant structures are demonstrated by the works of Antoni Gaudi, Sergio Musmeci, Heinz Isler, Pier Luigi Nervi, and Frei Otto [14,15], as well as research on physical models and the structural optimization of complex structures such as membranes, shells, and grid shells [16,17]. It is indeed possible to design novel structural systems that are structurally more effective by taking advantage of structural optimization and incorporating numerical simulations into a conceptual design [18,19]. Parametric structural design is the emerging research field that tries to bridge the gaps between current computational paradigms and enable true structural inputs into the conceptual design [20,21]. We may now create unprecedented levels of spatial and morphological complexity because of the development of computational tools and the associated formal-spatial repertoire [16]. Recent studies on the advantages of bio-inspired forms have already been linked with computational form-finding methods, it emphasizes the potential structural benefits of redundancy and differentiation, in addition to their ability to perform multiple functions at the same time [22,23]. Physical models have steadily been substituted in recent decades by algorithm-aided design because of the substantial advancements in Computer-Aided Design (CAD) and virtual modeling technologies, which enable more complicated, accurate analysis and computations [24,25]. The design problem can be approached using computationally developed techniques that pay close attention to both behaviorial requirements, architectural, and formal aspects. By using these techniques as design tools, they replace the physical models of earlier times with computational analysis, establishing a logical relationship between the architectural morphology and its structural stability [14,26,27].

In this study, we used a Genetic Algorithm (GA) and a Dynamic Relaxation (DR) algorithm to design a roof structure as a free-form shell with the aim of optimizing the shape. Structural shape and parametric models are created in the CAD tool Rhinoceros3D® with the aid of Non-Uniform Rational Basis Spline (NURBS) representation and associated with in-house built algorithms in Grasshopper® a plugin for Rhino, which was developed at Robert McNeel and Associates. Finally, FE models are produced to evaluate the structural performance of each design approach under applied loads. Results are discussed with reference to the structural engineer and designer, who must work together to identify the optimum architectural solution in the early design stages. We needed to integrate multiple digital tools to develop a computational process, including a CAD application, which can govern shapes parametrically, an optimization algorithm, and an FE solver. The workflow was developed entirely in the Rhinoceros3D®/Grasshopper® environment. Rhino's geometry is based on NURBS curves and surfaces, which can create free-form shapes. Grasshopper®, a built-in programmable plugin for Rhino, uses the graphic engine to present the results and enables an easy approach to parametric design and algorithmic modeling without the need for extensive scripting or programming skills, and it is also called a visual programming language. Grasshopper enables us to develop using generative algorithms that make use of associated modeling and generative modeling [24]. The interaction between CAD and the optimization process was made with available Grasshopper plugins that have GA and DR algorithms. For form-finding, dynamic relaxation solvers of Kangaroo Physics (plugin) [28], evolutionary solvers (genetic algorithm) of Galapagos [29], structural analysis of Karamba 3D (plugin) [30] in Grasshopper is used, and FE analyses are made possible with Dlubal RFEM during the design process. The interoperability between Dlubal RFEM and Rhino/Grasshopper is made with the RFEM common interface, an Application Programming Interface (API) that is provided by Dlubal software.

## 2. Method

The design development workflow is divided into two stages. First, we generated preliminary shapes in Rhino® using defined algorithms in Grasshopper® and performed the optimization. The spatial configuration of each individual was then transferred to the FE program, which analyzed the structure's stability. The conceptual design phase employs the exploration of GA and DR to conduct comprehensive form-finding analyses on the roof structure's shape. If the selected method does not meet the structural requirements, the ex-plorative process is restarted. The results that best satisfy functional and formal/aesthetic criteria are then used in the design evaluation stage to generate FE models and quickly check structural performance. Once a feasible solution is found that meets all the design criteria, then the detailed FE models are created to run the entire roof structure analysis (Figure 1).

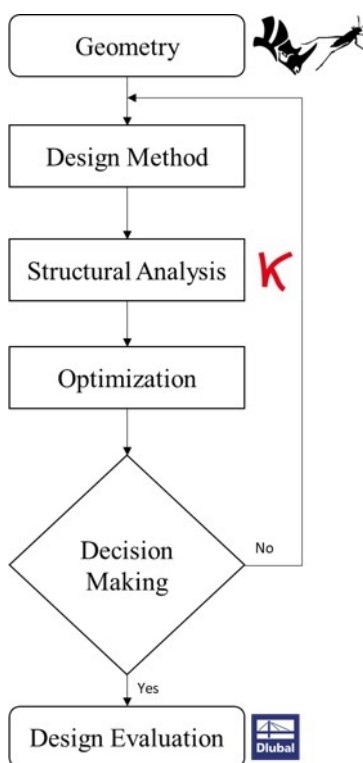

**Figure 1.** Design workflow of the application.

### 2.1. Structural Optimization

In general, optimization implies "improvement", but mathematically, it is a more precise concept: identifying the optimal solution by adjusting variables that are able to be controlled and commonly constrained. Optimization has widespread appeal as it is acceptable in all domains, and we can all identify with the desire to optimize things [31,32]. Any problem that requires a decision can be recast as an optimization problem. Depending on the optimization problem and model parameters, structural optimization can be classi-fied as shape, size, or topology [33–35]. The process of determining the optimal condition that gives the maximum or minimum value of an objective function $f(x)$ that is subjected to constraints and associated with its own design variables $x$ is known as structural opti-mization. Design variables are the parameters that can influence the structure to minimize (or maximize) different criteria [11]. Depending on the optimization problem, the design variables could be cross-sectional dimensions or member sizes, geometrical parameters, material properties, and nodal positions. The objective function in structural applications can be represented as the minimization of a structure's weight, volume, displacements,

stresses, or cost, so that a minimal-material structure can be found, and also maximization of stiffness in some cases [11,36].

The choice of an objective function is often one of the key decisions in the optimization process, and the classification of design variables must be carefully considered because it influences the success of optimization. In general, the mathematical formulation of any structural optimization problem is subject to constraints, including the values of design variables, and if a design satisfies all the requirements, then it is considered a feasible design [11]. Constraints are the limitations that must be met in order to create a feasible design. They are divided into two types: behavioral constraints and design constraints. Behavior constraints are derived from behavior requirements such as structural behavior, stress and deflections, which are called inequality and equality constraints. Design constraints, also known as side constraints, are based on several considerations such as functionality, aesthetics, buildability, and geometric and spatial requirements [11].

The mathematical formulation of a structural optimization problem can be expressed as follows. Minimization of an objective function $f(x)$ that is subjected to

$$
\begin{array}{lll}
g_j(x) \leq 0 & (=1, 2 \ldots, n_g) & \text{inequality constraints} \\
h_j(x) = 0 & (j = 1, 2 \ldots, n_h) & \text{equality constraints} \\
x_i^L \leq x_i \leq x_i^U & (i = 1, n) & \text{side constraints}
\end{array}
$$

where design variables are collected in the vector $\vec{x}^\top = [x_1, \ x_2, \ \ldots x_n]$.
In our case, we formulated a shape optimization problem in which the objective function is considered as total strain energy and design variables are subjected to side constraints.

$$
\begin{array}{ll}
\text{Objective Function:} & \text{Total Strain Energy} \\
\text{Design Variables :} & \text{"z" coordinates of the control points(NURBS surface)} \\
\text{Design Constraints :} & x_i^L \leq x_i \leq x_i^U
\end{array}
$$

### 2.2. Genetic Algorithm

GA is a heuristic method that uses Darwin's evolutionary theories to solve optimization problems. According to Goldberg [37], it is defined as a "search algorithm based on the mechanics of natural selection and natural genetics". It was developed at the University of Michigan by John Holland in the 1960s and formalized in 1975 [37]. It is considered an efficient tool for solving complex problems such as air traffic control, weather forecasting and many other areas where a combined analytic approach is unknown. The application uses evolutionary logic to solve specific problems. In genetic terms, the mathematical formulation can be formed by determining the representation scheme, the fitness measure, and design variables for controlling the algorithm and choosing the method of selecting the result and the termination criterion [37].

The representation scheme is a method of mapping the design variables, which includes defining a possible solution design space. Each variable of the problem is coded to become a part (gene) of a unique chromosome, which is the genetic representation of a possible solution (individual), and the range of variability of each gene is defined to limit the field of practical solutions exploration [18]. After the genetic formulation of the design variables, a suitable fitness measure is required. We measure fitness in the artificial world of mathematical algorithms in some way. This measurement is then used to control the operation that changes the structures in our artificial population. This value articulates the 'effectiveness' of generated solutions (individuals) relying on a well-defined control parameter and the associated evaluation criterion. As a result, a set of algorithm parameters should be specified. They are the population size, the maximum number of generations to perform, and the percentage of genetic operators used in conventional genetic algorithms (selection, crossover, mutations, etc.). Finally, when the existence of the best solution and thus best (or sub-optimal) reference performance values are known, a termination criterion should be defined. Otherwise, the algorithm can be stopped manually until a satisfactory solution is obtained [38,39]. Heuristic evolutionary solvers are used when optimization

problems have many variables, and an optimal solution cannot be found through exact solvers. In heuristic algorithms, there are many powerful methods available in the literature [8,12,13]. Nevertheless, we used GA because It is among the most effective methods for arranging the optimization problem for form-finding. All methods have disadvantages; GAs are extremely slow and do not guarantee a solution [40].

To initiate the process, the solver needs to populate the landscape or (model space) with a random collection of individuals or genomes, which is nothing but a specific value for each and every gene. There are two types of GA variants: binary and continuous. Each gene is assigned two values in binary (e.g., 0 or 1), and any continuous value can be used with upper and lower bounds in continuous variants.

In the case of binary GA.

$$X_i = \begin{cases} 1 & r_i < 0.5 \\ 0 & \text{otherwise} \end{cases} \tag{1}$$

In the case of continuous GA.

$$X_i = (ub_i - lb_i)r_i + lb_i \tag{2}$$

where $X_i$ is the $i$-th gene, $r_i$ is the random number in [0,1], and $ub_i$,$lb_i$ are the upper bound and lower bound for the $i$-th gene (variable).

The solver used in this study is Galapagos, a plugin for Grasshopper®, which is a single objective solver. The tool has two heuristic optimization algorithms, which are GA and Simulated Annealing. The Galapagos GA is slightly based on [41] using operators that only allow interpolation of similar solutions. To perform this process, GA requires five interlocking parts [29]:

- Fitness function.
- Selection.
- Mechanism.
- Coalescence Algorithm.
- Mutation factory.

For all optimization iterations, the Galapagos GA employs default parameters such as a maximum stagnant number of 50, a first boost multiplier of 2, a population size of 50, a maintained rate of 5%, and an inbreeding rate of 75% [29].

### 2.3. Dynamic Relaxation

The main objective of DR is to obtain a static equilibrium of a structure with material and parametric variations. Alistair Day invented DR in 1965 as a numerical procedure for solving a set of nonlinear equations. DR has been chosen as an alternative method because it enables us to form find the optimal shape within the equilibrium state. However, many other alternative methods are available in the literature [7,9,10]. Dynamic methods are some numerical techniques that describe the dynamic behavior of a system over a finite period by discretizing the continuum into sets of nodes with lumped masses that are interconnected via element or point masses called particles, regardless of the type of dynamic schemes and numerical integration used [41,42]. Each element is a topological subset of connected nodes through which forces are calculated. Hooke's law governs these calculations, which rely solely on the position of nodes. Typical well-established implementations of DR for form-finding have been developed to address the design of form-active structures with greater emphasis on tensile structures. In this sense, the method was entirely focused on the generation of equilibrium geometries with no shear or bending moments but with uniformly distributed in-plane tension or compression stresses.

For example, the gravity force over a lumped mass particle causes the displacement of the associated particle followed by the deformation of the connected springs. This deformation generates a counter force in the springs and elongates until the sum of the spring forces equals the mass's downward force [42]. Newton's second law of motion governs the particle's motion, and Hooke's law of elasticity governs the force in the spring. Newton's second law governs the motion of each node and can be decoupled into a translational and a rotational part [42,43]. To calculate translational motions, each node has variables for describing its position, velocity and residual force that are stored through three-dimensional vectors. The nodal mass for each coordinate direction can also be described through individual values but, for simplicity, a single scalar value $M$ is normally used for all directions. Conceptually, a node moves through three-dimensional space with a certain velocity caused by a constant acceleration that is derived from all forces acting on the node at a certain point in time.

$$\vec{x} = \begin{bmatrix} x_1 \\ x_2 \\ x_3 \end{bmatrix} \quad \vec{v} = \begin{bmatrix} v_1 \\ v_2 \\ v_3 \end{bmatrix} \quad \vec{F} = \begin{bmatrix} F_1 \\ F_2 \\ F_3 \end{bmatrix} \tag{3}$$

In addition to this, variables for orientations, angular velocities, residual torques, and moment of inertia may be included to calculate rotational motions. In the discretized system, the translational motion of a node based on Newtonian dynamics is governed by

$$\vec{F}_i^t = M_i^t \cdot \vec{a}_i^t \tag{4}$$

where $\vec{F}_i^t$ is denoted by the translational residual force, $M_i^t$ is the lumped mass, and $\vec{a}_i^t$ is the acceleration of the node in the global coordinate system at the time.

To ensure numerical stability, the lumped mass of a node needs to be adjusted at each time step based on the greatest direct stiffness $S_i^t$ calculated on a node.

$$M_i^t = \frac{\Delta t^2}{2} \cdot S_i^t$$

The equivalent rotational motion of a node based on Newton's second law of motion is governed by:

$$\vec{T}_i^t = \vec{I}_i^t \cdot \vec{\alpha}_i^t + \vec{\omega}_i^t \times (\vec{I}_i^t \cdot \vec{\omega}_i^t)$$

where $\vec{T}_i^t$ is denoted by the resultant residual torque, $I_i^t$ is the moment of inertia and $\vec{\alpha}_i^t$ is the angular acceleration of the node in the global coordinate system at the time.

For form-finding and analysis, the behavior of any structure is modeled through linear elements such as discrete network assembly, in which each link is associated with an elastic bar element permitting the calculation of forces. For example, the tension $T_m$ of an elastic bar $m$ connecting a node $i$ with an adjacent node $j$ at a time is calculated by:

$$T_m^t = T_m + K_m \cdot (L_m^t - L_m) \tag{5}$$

where $L_m^t$ is the current length of the element, and $L_m$ is its initial length.

In form-finding, the factors $T_m$ and $T_m$ are the respective geometric and elastic stiffnesses of the elastic bar that are used by designers to control the equilibrium geometry and the distribution of pre-stress [43]. In the case of load analysis, $T_m$ is the tension derived from the form-found geometry, and the elastic stiffness is defined by $K_m = EA/L_m$, where $E$ is Young's modulus and $A$ the cross-sectional area of the element. On this basis, the internal axial force of the elastic bar connecting to is resolved by:

$$\vec{f_m,i}^t = \frac{\vec{T}_m^t}{L_m^t} \cdot (\vec{x}_j^t - \vec{x}_i^t) \tag{6}$$

Hence, the residual force acting on the same node is calculated by adding the axial forces of all the connected elastic bars and external forces as:

$$\vec{F}_i^t = \sum_m (\vec{f_m i})^t + (\vec{f}_{ext\,i})^t \tag{7}$$

To define the nodal mass $M_i^t$ of the node, the greatest direct stiffness is obtained by:

$$F_i^t = \sum_m \left( \frac{EA}{L_m} + g \cdot \frac{T_m}{L_m^t} \right)$$

where $g$ is a constant factor that may be used to calibrate the contribution of the geometric stiffness when defining the nodal mass.

Kangaroo, a Grasshopper® plugin, was used to create this form-finding framework. The Kangaroo is a set of algorithms that allows for the simulation of the behavior of the physical object. It is also known as the Physic engine for Grasshopper®, and the simulations are dynamic [28]. To arrange the physics engine, we need to define the goal objects: anchorage points, length line and load.

1. Anchor: This maintains a point's original position, and it is used to hold the mesh's corners in place.
2. Length (Line): This goal object aims to maintain two points (line endpoints) at a specific length apart. The points move less when the strength is increased.
3. Load: This goal object applies a normal force on the points. The force is specified as a vector, and the length of the vector is proportional to the magnitude of the applied load.

Finally, the Kangaroo Solver collects all the goal objects and solves the system.

## 3. Application

The canopy of the ecological Island of AMA Roma (Municipal Urban Hygiene Company) has been conceived for design development using optimization algorithms (Figure 2): The Ecological islands are facilities open to citizens' equipment for the temporary holding of non-organic waste, arranged in a unique way, to be followed by recycling activities [44,45]. The new canopy design had to meet functional needs, such as accommodating vehicles for loading/unloading operations or minimizing the number of supports to facilitate vehicle travel while considering the differentiated height of supports. Because the existing canopy was too thick and rigid for this reason, we would like to present a new feasible hypothesis of an approach that allows us to assume a realizable form, which was explored using an optimization procedure in the CAD software Rhinoceros®/Grasshopper®.

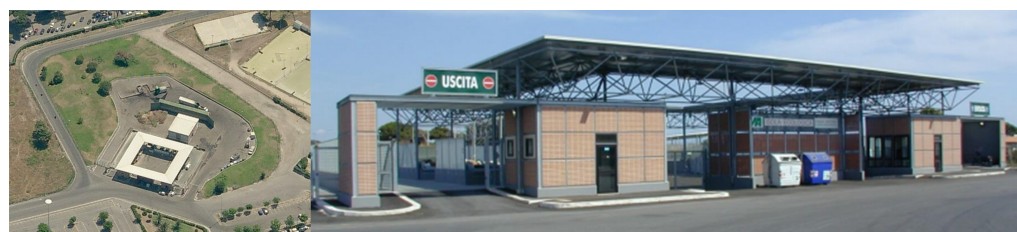

**Figure 2.** Ecological island, waste collection center, Rome, Italy.

The original measurements of the canopy were a 33 × 33 m steel truss structure with an accessible area inside, with 22 columns. The architectural concept of the new canopy design was inspired by the up-cycling logo, which stands for reuse rather than recycling. The canopy was modeled after an outline of the logo, and the final shape was achieved by exploiting the NURBS curve control points (Figure 3).

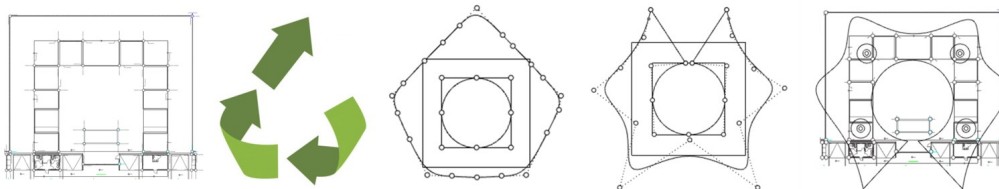

**Figure 3.** The architectural concept of the roof cover.

The spatial representation of the canopy has been described in Rhinoceros 3D® software using a three-degree NURBS surface with a web of $4 \times 21$ interpolating points (Figure 4a), with the vertical coordinates of those points assumed as design variables. The maximum vertical displacement of the roof shape was chosen as a structural parameter to assess its local mechanical behavior. Similar results could be achieved by deriving the fitness function from certain integral parameters of the structure, such as its total strain energy, which represents a global measure of its mechanical behavior. The shape of the surface is then formulated as a shape optimization problem to obtain a free-form shell with respect to minimizing the total strain energy across the shell surface under self-weight. In this case, design constraints are considered a functionality, limiting the range of slopes.

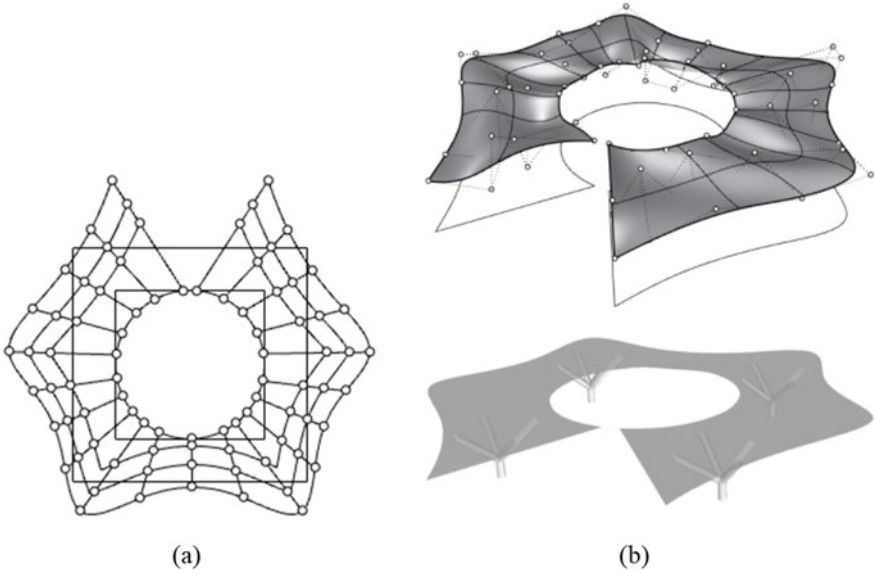

(a)  (b)

**Figure 4.** (**a**) NURBS control points representation of the roof cover. (**b**) NURBS surface and column positions of the roof cover.

The NURBS surface control points are imported into Grasshopper in order to arrange the optimization problem. First, the control points are deconstructed to make vertical coordinates a design variable, where $x$ and $y$ coordinates are fixed in the same position, and $z$ coordinates are parameterized. It is a parametric representation of a geometric entity that could be considered analogous to the Genetic Algorithm (genotype).

In this case, the canopy's maximum height has been determined in relation to its functionality, and a minimum height of 5.6 m must be guaranteed to accommodate the vehicles for loading/unloading operational activities. The location and height of the columns, along with the plan projection of the roof boundary and its thickness of 15 cm, have therefore been identified as boundary conditions. To this aim, the constraints for the optimization problem are defined by design constraints considering the functionality and values that can be used with upper and lower bounds.

In the shape optimization method, it is well known that the functioning of the optimization process largely depends on the definition of the range of variability of each variable; that is, a section of its boundary can be expressed as a search space or a domain

range, and an optimal shape for this region is sought based on a specified criterion. Therefore, the domain range between upper and lower bounds is provided with 2 m for design variables to find an optimal shape. Total strain energy must be determined after defining the domain range to define the objective function and arrange the shape optimization problem. To this aim, we used a plugin called Karamba 3D, which is an interactive, parametric finite element program developed by Clemens Preisinger in cooperation with Bollinger and Grohmann ZT GmbH Vienna, Austria [30]. It enables us to integrate parametric models with finite element calculations and optimization algorithms, which helps us to search for optimal shapes by using algorithms such as Galapagos. A surface has been created from interpolating points and transformed into a mesh, which is of size (50 × 50), as in Karamba3D, shells are based on meshes. A tree column with four branches at each corner of the canopy served as the support. Thus, the columns are reduced from 22 to 16 (4 columns × 4 branches) (Figure 4b). At each column that connects to the ground, the nearest four points from the defined mesh vertices are randomly selected to define the support system.

The mesh's vertices are subjected to a 2 kN nodal force in the negative $z$-direction, and one gravity load has been coupled to create a load case. The Karamba calculates the deflection for each loading scenario, adds this information to the model, and outputs the maximum nodal displacement and the maximum total force of gravity, and for information on work and energy, we considered the internal deformation energy of each load situation for the structure. These results are utilized to rank the outcome during a structural optimization procedure: the smaller the maximum deflection, the amount of material used, and the value of internal elastic energy, the more efficient the structure. Finally, the Evolutionary Solver Galapagos requires two inputs to solve the shape optimization problem and perform the genetic algorithm: genome and fitness. A fitness value is defined as the value to be optimized; in this case, it is total strain energy, which is the objective function. The genome is a collection of parameters that influence fitness; in this case, it is the vertical coordinates of the NURBS surface control points ($z$ coordinates), which are the design variables.

The algorithm will find the shape of the shell by changing the height of the $z$ coordinates of the NURBS control point within the given range or domain; if there is an increment in the strain energy by the transformation at node 'i', the algorithm pulls down the $z$ coordinate, and if in the case of a decrease in the strain energy by the transformation at node 'i', the algorithm pushes up its z coordinate. This method produces a curved surface shape with reasonable structural rationality while maintaining and satisfying functionality and visual requirements. The shape was optimized by means of a GA, the solver gave numerous solutions (Figure 5) and the best-performed shape was selected (Figure 6). Regrettably, the evaluation of the GA outcome did not meet the structural requirements. However, design iteration was not considered because the purpose of this application was not related to tool efficiency but rather to structural efficiency experimentation. The primary issue that arose is geometrical form, which is vitally pertinent in architectural and engineering problems. The shell's shape was quite inflated around the corners. There is a translation from the external boundary's NURBS curve to the circular curve of internal space, but the interpolation from the NURBS curve to a circle is problematic, and the problem was the structural complexity of the shape, which would be difficult to fabricate.

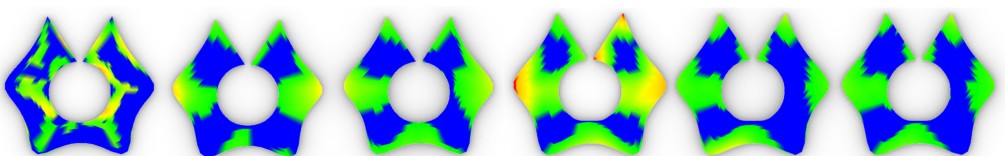

**Figure 5.** GA outcome.

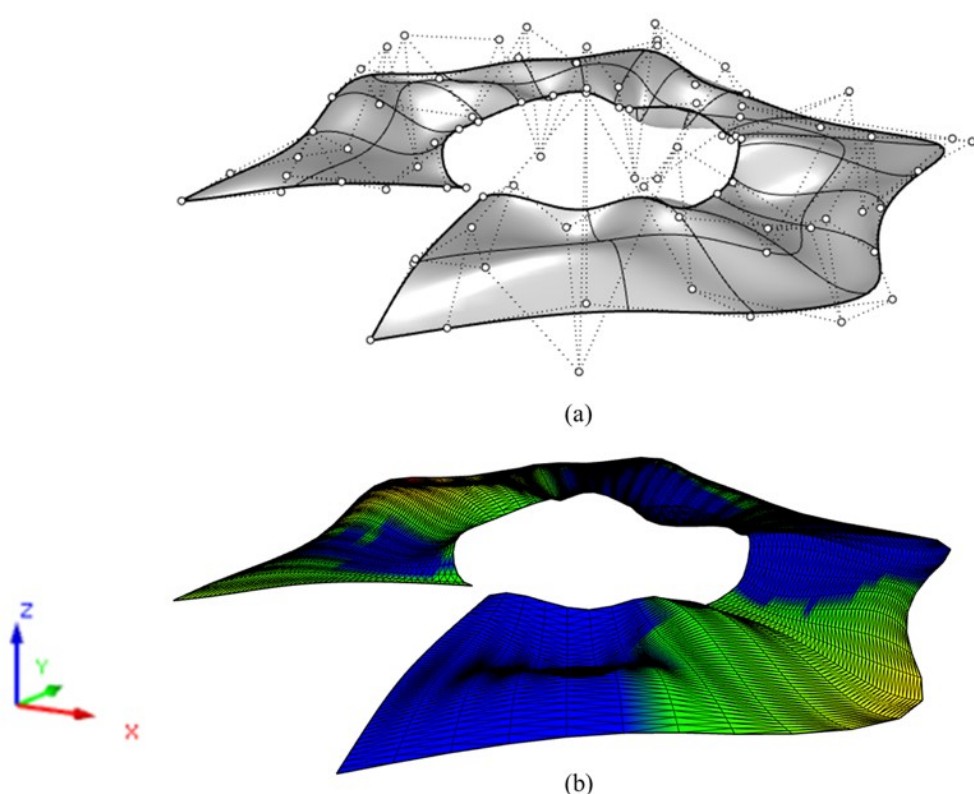

**Figure 6.** (**a**) Optimized NURBS surface with displaced control points. (**b**) GA's best outcome of the roof cover.

To address this issue, two approaches can be considered: first, changing the continuous shell to a grid shell, and second, changing the design approach. To achieve this goal, we first decided to switch from a continuous shell to a lattice shell, which combines the aesthetic of form with the performance of a structure driven by the construction simplicity and material economy of straight lath members that can be bent into shape. As a result, a Kagome lattice (Aniso grid) [46,47] made up of triangles and hexagons was chosen to project on the free-form surface generated by the evolutionary solver (Figure 7). The lattice structural grid was developed using the Lunch Box plugin in Grasshopper. The Lunch Box plugin in Grasshopper was used to create the lattice structural grid. A free-form surface is paneled with hexagonal cells in this process. It generates a flat list of hex cells, and by connecting the midpoints of the hexagon sides, we obtain a Kagome lattice grid.

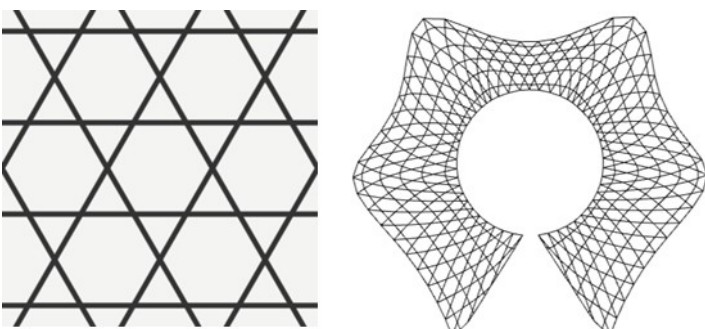

**Figure 7.** Kagome lattice grid pattern.

Aniso grid structures are not limited to the equilateral triangle and can be consistent with a variety of patterns. In other words, their structural patterns may be non-isotropic. To run the entire Finite Element Analysis, we used the structural analysis program Dlubal

RFEM, which is a powerful 3D FEA program that is based on a modular software system and includes a default interface to Rhino and Grasshopper. RhinoRfem is an open-source plugin that connects RFEM and Grasshopper via the Dlubal RS-COM interface. Because the structure is symmetric, we only examined half of it to reduce computation time (Figure 8). Regrettably, the structural performance was insufficient to withstand the load transmitting on the canopy. As a result, the design solution was not considered.

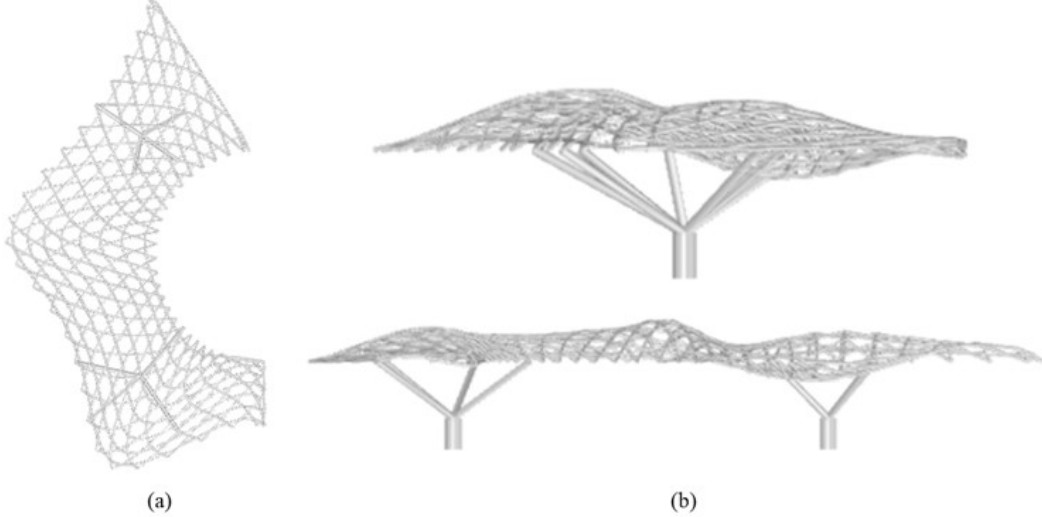

**Figure 8.** The resulting roof cover by considering a grid shell (**a**) Top view. (**b**) Side view.

As previously stated, if the selected design approach does not meet structural requirements, the exploratory process is restarted. As a result, we decided to alter our design strategy. To find the best form, a dynamic relaxation method was chosen. The form-finding process was integrated into this workflow to achieve this goal by introducing the Kangaroo [28]. The shape of the roof was created by defining a single surface and then relaxing a grid over it. The design workflow begins with defining the canopy's boundary NURBS curves. The boundary curves are defined manually in Rhino®, and the objective goals are defined in Grasshopper®. The Kangaroo solver then gathers all the goal objects, which are anchor points, spring length and load. Finally, it solves the roof system, and the solution established the boundary condition using a catenary control rig (a node-based rigging system), from which the surface was formed using Kangaroo and a network of springs that balanced spring length equalization (surface relaxation) with an upwards load vector (inverse hanging-chain model). It returns the "solved" vertex locations; by combining those vertex points, we can generate a mesh. Finally, the mesh was transformed into a surface, and a flat Kagome pattern was projected onto the resulting free-form surface (Figure 9).

Finding the optimal shape for a canopy for desired or required constraints and goals in the form-finding procedure described above is essentially an optimization process, usually followed by trial and error. The structural grid went through several stages before reaching its final form. The grid meets the ground with four support systems in the early scheme, which was structurally unstable and unsatisfactory, leading to six support systems after many iterations and progressive changes in the number of support systems and the geometry's shape (Figure 10).

The model was exported to RFEM to perform the FE analysis and evaluate the performance, with a hot-finished Steel S 355 circular pipe of 100 mm diameter and 10 mm thickness, Young's modulus $E = 210,000$ MPa, and a load of 1 kN applied to all intersecting nodes of the grid structure in the negative $z$ direction. The foot nodes of the columns were assigned nodal support, with six nodal supports at each supporting system. A 2 mm thick tensile membrane (Polytetrafluoroethylene) fabric sheet was chosen to stretch over the lattice grid as a cover for the roof structure (Figure 11).

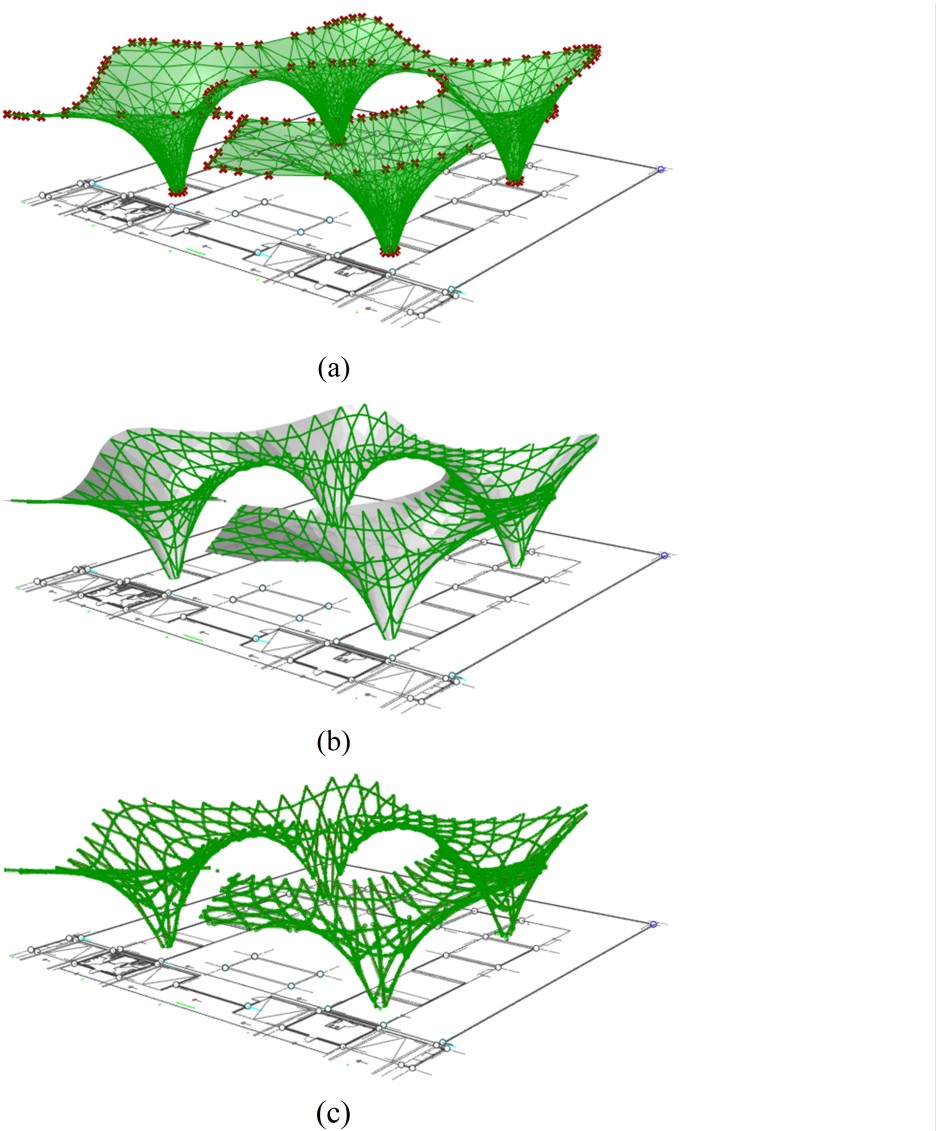

(a)

(b)

(c)

**Figure 9.** Formation of grid shell through surface relaxation. (**a**) Mesh relaxation from anchorage points. (**b**) Kagome grid pattern projection on a mesh. (**c**) Final grid shell.

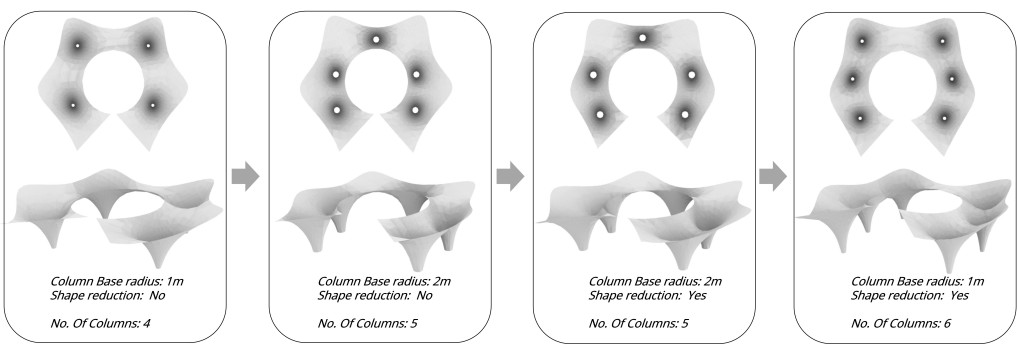

**Figure 10.** Evolution of form-finding process of the roof cover.

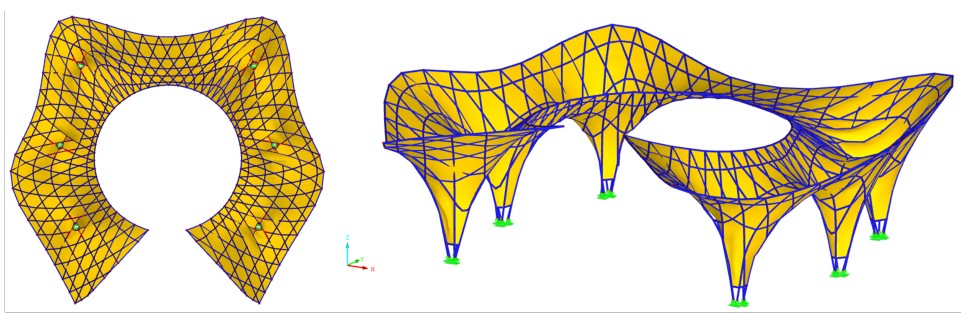

**Figure 11.** FE model to evaluate in Dlubal RFEM.

## 4. Results

Once all material properties, supports and loads are defined, the model is ready to analyze. RFEM, FE solver calculations are based on second-order analysis as our structural model is nonlinear, and RFEM automatically decides the required number of load increments according to a heuristic method. The calculation included wind load cases and snow load cases, in addition to the self-weight, and the load situations were provided by the Dlubal software online GEO-ZONE tool. The structural analysis of beam members and the tensile surface was performed by the FE solver, and results were obtained with a maximum displacement of 167.4 mm for beams (Figure 12) and 57.2 mm for surfaces (Figure 13).

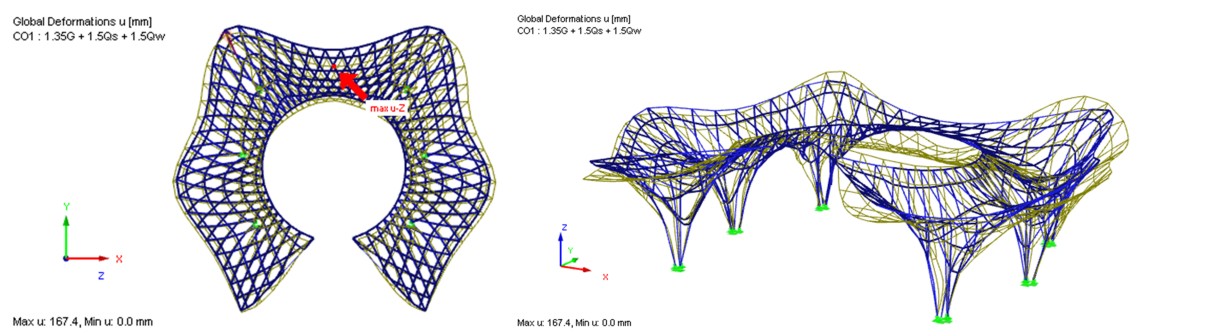

**Figure 12.** Finite element analysis of the structure made of beam elements.

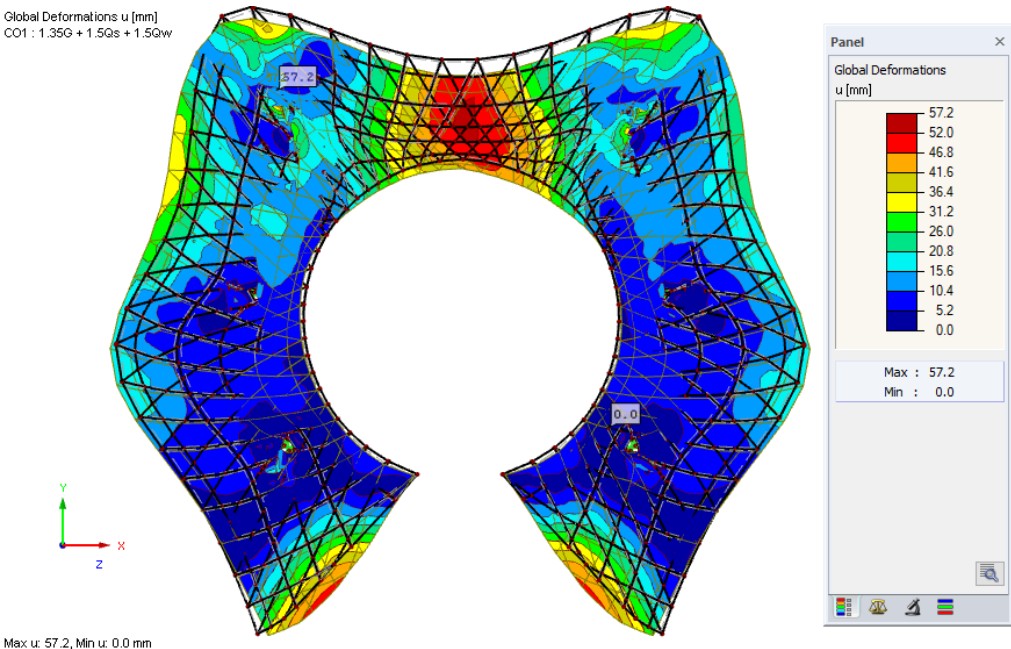

**Figure 13.** Finite element analysis of the roof surface.

## 5. Conclusions

In this study, a free-form shape-resistant roof structure has been designed with the aid of a GA and DR. Finally, FEM analysis is performed for each approach to verify and validate the structural capacity. However, the development and application of these complex interactive and iterative design approaches through computational tools should only be considered as a starting point for a more general discussion on the role played by new digital technologies, specifically in the construction world. The optimal results demonstrate that the optimization process based on the Relaxation Algorithm converges faster than the Genetic Algorithm but also provides some insights into how both algorithms explore the feasible solution domain. The optimization path is normally influenced by the design parameters, particularly how the search domain is adjusted during the optimization process to enhance computation time. As a result, the development of GA efficiency has not been considered in this application. Algorithms can even be considered generators of unexpected optimal or suboptimal shapes in free-form optimization problems. This feature enables designers to use them as form-finding tools in a creative manner, involving other aspects of the design process aside from the solution of the planarity problem, such as aesthetics or formal innovation. The creation and improvement of good sub-optimal solutions during the iterative procedure, as well as the ability to interact with the procedure itself during the design process, allow designers, both architects and engineers, to develop innovative projects while maintaining control of the performance.

In conclusion, the defined methodology is suitable for the rapid exploration of design concepts with a qualitative understanding of structural behavior while establishing an awareness of the downstream impacts of design operations and facilitating a sensible interaction between aesthetic quality and structural performance, as well as CAD geometries. This enables us to understand how a parametric structural design framework can facilitate the decision maker to generate and analyze the optimal design solutions rapidly in the early stage of the design process. The Rhino RFEM common interface API provided by Dlubal was a better choice for this preliminary research, integrating CAD models and the FE analysis program, making this design workflow seamless. The design approaches are adaptable and have been effectively implemented to the roof structure under a wide range of constraints. An important aspect of developing proper constraints is careful consideration of how the structure will be constructed and fabricated. Although the resulting discrete wireframe model must still be dimensioned and materialized, evaluating structural performance using FE analysis could significantly improve the conceptual design. This study clearly demonstrates the benefits of Structural Optimization by incorporating Finite Element Analysis into the conceptual design in order to create innovative structural systems that are structurally more efficient. However, this preliminary investigation provided a few valuable insights into the opportunities for enhancing the parametric structural design, and additional research is required to consider various structural and architectural requirements.

**Author Contributions:** S.K.D.: Conceptualization, Methodology, Software, Investigation, Formal Analysis, Data Curation, Validation, Writing—Original Draft, Writing—Review & Editing, Visualization. N.F.: Supervision, Conceptualization, Methodology, Software, Validation, Writing—Review & Editing. P.T.: Supervision, Funding acquisition, Writing—Review & Editing. R.P.: Supervision, Conceptualization, Methodology, Validation, Writing—Review & Editing. M.P.: Conceptualization, Methodology, Validation, Writing—Review & Editing. All authors have read and agreed to the published version of the manuscript.

**Funding:** This research has been partially supported by Sapienza University of Rome funds (PhD Program in Structural and Geotechnical Engineering, University Starting Grant 2022).

**Data Availability Statement:** The data presented in this study are available on request from the corresponding author.

**Conflicts of Interest:** The authors declare no conflict of interest.

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
