# Peer review of "Optimal Design of a Canopy Using Parametric Structural Design and a Genetic Algorithm"

_symmetry, doi:10.3390/sym15010142_

Round 1

Reviewer 1 Report

Using a Genetic Algorithm (GA) and a Dynamic Relaxation (DR) Algorithm, a framework structure was created in this article as a free-form shell with the goal of optimizing the shape. In order to achieve this, fully parametric models were built using Rhinoceros3D's NURBS representation in conjunction with an internal genetic algorithm and relaxation algorithm. Finite Element (FE) models were then generated for each design approach in order to assess structural performance. under a specific load scenario.

The authors combine several digital tools to create a computational workflow, including an optimization algorithm, a FE (Finite Element) solver, and a Computer-Aided Design (CAD) application, which can offer parametric control over shapes. The literature on optimization algorithms, FE (Finite Element) method, and Computer Aided Geometry Design is replete with commercial and open-source computer programs, articles, and books. The original point of this work is the development of a methodology that integrates these three tools. The weak part of the article was that very little information was given about how this integration was done. As is known, optimization is an iterative process. More detailed information can be given about how these three programs were integrated to work automatically under a single program.

Author Response

Thanks for your insightful comments. We have now included the following clarifying statement in introduction.

Line (83)

The interaction between CAD and the optimization process was made with available Grasshopper plugins that have GA and DR algorithms.

Line (88)

The interoperability between Dlubal RFEM and Rhino/Grasshopper was made with RFEM common interface which is an Application Programming Interface (API) that is provided by Dlubal software.

Author Response

Reviewer #2

The paper is attractive but the answers for the following questions should be added before its publication:

  1. Why DR is used in place of matrix methods and finite element for the analysis, e.g. see the following book?
  2. Kaveh, Computational Structural Analysis and Finite Element Methods, Springer Verlag, Springer International Publishing, Switzerland, 2014.

Reply to Reviewer Comment

Thanks for your insightful comments. Well, we used DR to have a smooth shell structure that is in static equilibrium state. However, thanks for your suggestions we would like to implement those methods in the future work. We have now included a following statement in the manuscript.

Line (180)

DR has been chosen as an alternative method because it enables us to form find the optimal shape within the equilibrium state. However, many other alternative methods are available in the literature [A.Kaveh, 2014].

  1. Why Genetic algorithm is used in place of powerful recent methods. Twenty five such methods are explained in the following books:
  2. Kaveh, Advances in Metaheuristic Algorithms for Optimal Design of Structures, Springer International Publishing, Switzerland, 3rd edition 2021.
    A. Kaveh and T. Bakhshpoori, Metaheuristics: Outlines, MATLAB Codes and Examples, Springer, Switzerland, 2019.

Reply to Reviewer Comment

Thanks for your suggestions. Well, we used GA because that allows us to design alternate solutions before finalizing the solution and it is one of the most efficient approaches for form finding. However, we find the resources that you provided are very insightful and we are interested in implementing in a future work. We have now included the following statement in the manuscript.

Line (155)

In heuristic algorithms, they are many powerful methods available in the literature [A.Kaveh, 2019,2020]. Nevertheless, we used GA because it is one of the most efficient methods for the arrangement of the optimization problem of form-finding.

  1. Why graph product, canonical forms are not used in place of utilized approach? Many good methods are available in the following books:

A.Kaveh, H. Rahami and I. Shojaee,
Swift analysis of structures using graph theory methods, Springer, 2020,
A.Kaveh, Optimal Analysis of Structures by Concepts of Symmetry and Regularity, Springer Verlag, GmbH, Wien-NewYork, 2013.
A. Kaveh, Topological Transformations for Efficient Structural Analysis, Springer, 2022.

Reply to Reviewer Comment

Thanks for your suggestions. Suggested references have been cited accordingly in the revised version of the manuscript. We have used more available methods that can be utilized within the described framework. However, we would like to implement other methods you mentioned in our future works.

Reviewer 3 Report

Comments:

In this work, the authors successfully integrated the FEM into design optimization software to solve a practical problem. The overall flow of the manuscript is well and it will help the researchers  regarding how to utilize available software for design optimization problems.

However there are few things to be addressed in the revision.

1.     This is completely an application based work and the authors should state the novelty precisely in the abstract and in the conclusion. (compactly in a single or by a couple of sentences)

2.     There are some typos throughout the manuscript. Please check and correct them.

3.     Write the unit correctly (33 X 33 m : Line 212)

4.     What is K (in red) in fig. 1. It seems like written in paint and inserted manually. The borders of K is visible. Also, mention what is K in the manuscript.

5.     All the figures are not referred in the manuscript. Please refer them in the text and explain them.

6.     Most of the Figure’s captions are very general.

Explain all the figures in their caption so that the reader can understand it properly. For example in Fig. 6, what are the nodes in the top figure and what are the explanations for the colors in the below figure. What the colors are representing. These are not clear at all. Also, if there are more than one figure mention them as a,b,c… and explain them on the manuscript.

7.     Resolution of some figures are not good and the texts inside the figures are not quite readable (for e.g figure 10). Correct them in the revision.

Author Response

Reviewer #3

In this work, the authors successfully integrated the FEM into design optimization software to solve a practical problem. The overall flow of the manuscript is well and it will help the researchers regarding how to utilize available software for design optimization problems.

However there are few things to be addressed in the revision.

  1. This is completely an application based work and the authors should state the novelty precisely in the abstract and in the conclusion. (compactly in a single or by a couple of sentences)

Reply to Reviewer Comment

Thanks for your insightful comments. We have now included the following clarifying sentence in abstract and conclusion.

Line (6)

We demonstrate how parametric structural design can facilitate the decision maker to generate and analyze the optimal design solutions rapidly in the early stage of the design process.

Line (378)

This enables us to understand how parametric structural design can facilitate the decision maker to generate and analyze the optimal design solutions rapidly in the early stage of the design process.

  1. There are some typos throughout the manuscript. Please check and correct them.

Reply to Reviewer Comment

We have now revised all the typos throughout the manuscript.

  1. Write the unit correctly (33 X 33 m : Line 212)

Reply to Reviewer Comment

We have now revised.

  1. What is K (in red) in fig. 1. It seems like written in paint and inserted manually. The borders of K is visible. Also, mention what is K in the manuscript.

Reply to Reviewer Comment

Well, the K is the logo of the software Karamba 3D that used during the application process. We have now modified the figure 1 and mentioned it in the manuscript.

  1. All the figures are not referred in the manuscript. Please refer them in the text and explain them.

Reply to Reviewer Comment

Thanks for your kind remainder. We have now referred all the figures in the manuscript.

  1. Most of the Figure’s captions are very general.

Explain all the figures in their caption so that the reader can understand it properly. For example in Fig. 6, what are the nodes in the top figure and what are the explanations for the colors in the below figure. What the colors are representing. These are not clear at all. Also, if there are more than one figure mention them as a,b,c… and explain them on the manuscript.

Reply to Reviewer Comment

We have now revised the figures and their captions with explanation in manuscript.

  1. Resolution of some figures are not good and the texts inside the figures are not quite readable (for e.g figure 10). Correct them in the revision.

Reply to Reviewer Comment

We have now revised Figure 10.
